# The Rate of Hospitalization of Pregnant Women with Multiple Sclerosis in Poland

**DOI:** 10.3390/jcm11195615

**Published:** 2022-09-23

**Authors:** Dorota Walkiewicz, Bożena Adamczyk, Michał Maluchnik, Jakub Perwieniec, Krzysztof Podwójcic, Mateusz Szeląg, Michał Zakrzewski, Konrad Rejdak, Agnieszka Słowik, Marcin Wnuk, Monika Adamczyk-Sowa

**Affiliations:** 1Department of Analysis and Strategy, Ministry of Health in Poland, 00-952 Warsaw, Poland; 2Department of Neurology in Zabrze, Faculty of Medical Sciences in Zabrze, Medical University of Silesia, 40-055 Katowice, Poland; 3Department of Adult Neurology, Medical University of Gdansk, 80-211 Gdansk, Poland; 4Institute of Labour and Social Studies, 01-022 Warsaw, Poland; 5Department of Neurology, Medical University of Lublin, 20-059 Lublin, Poland; 6Department of Neurology, Jagiellonian University Medical College, 31-008 Krakow, Poland; 7Department of Neurology, University Hospital in Krakow, 30-688 Krakow, Poland

**Keywords:** multiple sclerosis, pregnancy, administrative data, Poland

## Abstract

Multiple sclerosis (MS) is most often diagnosed in women of childbearing age. Therefore, it is important to examine the impact of pregnancy on the course of MS and to enable patients to make decisions about motherhood based on reliable data. The main objective of this study was to assess the impact of pregnancy on the course of MS by comparing the frequency of MS-related hospitalizations during pregnancy and 40 weeks postpartum versus 40 weeks before pregnancy. We used administrative health claims to identify female patients with MS, their deliveries, and their MS-related hospital admissions and calculated the frequency of MS-related hospital admissions before, during, and after pregnancy. We observed that MS is diagnosed approximately three times less often during pregnancy than before or after pregnancy. The number of MS-related hospital admissions decreased during pregnancy, especially in the third trimester. In contrast with other studies, we did not observe an increased level of MS-related admissions postpartum. The number of hospitalizations reported with steroid injections and emergency department visits also decreased during pregnancy. Our results show that pregnancy has a protective effect on the course of MS.

## 1. Introduction

Multiple sclerosis (MS) is a chronic, inflammatory demyelinating disease of the central nervous system with multifocal damage to the nervous tissue. More than 70% of patients are women and the peak incidence occurs between the ages of 20 and 40, which is the period of the highest reproductive activity. Therefore, the issue of the influence of MS on the course of pregnancy as well as the health of the mother and the newborn has aroused great interest over the years due to the practical implications of family planning and treatment of patients with MS.

The latest reports published by Bonavita et al., revealed that people with MS more often than the general population do not have children, especially in the group of patients aged 36–45 years [1]. Moreover, in this study, 56% of respondents with MS reported that the disease to a varying degree influenced family planning decisions. In total, 14% of participants refused to have children completely. It is a much smaller percentage than noticed in older studies [2]. Increasing knowledge about pregnancy in MS may have an impact on the making decision about having children.

That is why it is so important to explore the topic of MS in the context of pregnancy so that young women faced with the need to make decisions about motherhood receive reliable and verified information.

The special hormonal environment during pregnancy favored the Th2 lymphocyte response, which could block the progression of inflammation associated with MS [3]. Preliminary observations suggest that oral contraceptives may be protective, and that differences in the rate of progression may depend on genetic factors related not only to sex but also to sex hormones [3].

There is a difference in disease activity during pregnancy (and also before and after childbirth). For example, in MS and rheumatoid arthritis, the severity of the disease decreased during pregnancy, backed to pre-pregnancy levels after birth [3].

As research shows among women with MS, the frequency of relapses during pregnancy decreases (0.7 ± 0.9 before pregnancy vs. the third trimester of pregnancy, where it is 0.2 ± 1.0) and increases in the first three months after delivery to 1.2 ± 2, 0, and then goes back to the pre-pregnancy rate [4]. Despite these data, the current pregnancy rate in the population of women with MS is increasing [5].

However, the influence of pregnancy on MS is also of great importance among patients with an aggressive form of MS, who often give up parentage for fear of disease progression. Recent data indicate that women with highly active MS may need DMT maintenance until pregnancy is confirmed, and sometimes throughout pregnancy, to avoid relapse of disease activity and severe relapses during pregnancy [6].

Nevertheless, it should be remembered that high disease activity may be one of the main reasons for not having children in MS patients. A total of 71.2% study participants believed that the symptoms of MS could disrupt parenting [1].

Among women with MS in the last trimester of pregnancy, the level of estrogen in the blood was at the highest level, which correlated with a reduction in the frequency of relapses. When considering the protective effect of estrogens in pregnancy, it seems warranted to investigate in the future whether exogenous estrogens could provide protection against disease progression [7].

Moreover, there are reports that estradiol administered in the physiological dose that is secreted during pregnancy has a protective effect and counteracts radiological progression [8,9,10]. Due to the numerous risks of complications, including cancer and thromboembolism, such treatment is not common but it confirms that pregnancy is a period of silencing the progression of MS. It seems that it is more beneficial for women diagnosed with MS to make a decision about the pregnancy earlier in life. Unfortunately, patients may postpone the decision to become pregnant and give birth at a later age due to relapses and the need for treatment.

The main aims of this study were: (1) to compare the frequency of MS-related admissions in the period before, during and after pregnancy; (2) to assess how often the first diagnosis of MS is made during the pregnancy compared with the period before and after pregnancy; and (3) to assess the level of participation in DMT among pregnant women with MS. In contrast to most studies to date exploring the relationship between pregnancy and MS, this study is a retrospective analysis of the most current as possible administrative data combined from several sources. As a result, we were able to trace several years of the medical history of a large cohort of Polish MS patients, who had access to the most advanced treatment for MS offered in the last decade.

## 2. Materials and Methods

We used electronic administrative health claims collected from 2009 to 2019 by the National Health Fund (NFZ), a single public health care payer in Poland, and data from the Institute of Mother and Child (IMC) collected from 2011 to 2019 for the purpose of screening newborns. The NFZ database comprises individually reported data claimed to the payer by service providers. The data include detailed service description and demographic variables describing the patient. The IMC database comprises individually reported childbirths with the exact date of childbirth and pregnancy duration in weeks.

The data collected in both these databases are anonymous but individual patients can be distinguished and matched by their IDs, which are pseudonymized national identification numbers. Deliveries reported to NFZ and IMC overlap by more than 95%. The main two reasons for incomplete overlap are childbirths in the private sector (in such cases births are reported to IMC but not NFZ) and reporting errors, especially in the patient’s ID.

In Poland, there is a national registry of MS patients does not exist. In this study, MS patients were identified on the basis of NFZ data using the following criteria: participation in disease-modifying therapies (DMT) for MS patients and at least three reported visits with “G35” ICD-10 code (denoting multiple sclerosis according to ICD-10 classification) between 2009 and 2019. Overall, 13,358 female patients met these criteria. The date of the first service with the “G35” ICD-10 code reported in the NFZ database was considered the first MS diagnosis. Therefore, it is possible that the patients considered in our analysis as “diagnosed with MS” at the moment of delivery may not fulfill the McDonald criteria yet.

In the next step, a group of women with delivery in medical history has been extracted from the cohort of female MS patients (Figure 1). Deliveries were recognized by specific diagnosis-related group (DRG) products reported for the service in the NFZ database and then matched with the IMC data by patients’ ID and date of childbirth. Overall, 2876 childbirths in 2011–2019 were identified (Table 1), including deliveries of patients before or after the first MS diagnosis. Only pregnancies that ended with deliveries between 2011 and 2019 were included in the further analysis because the database collects data since 2009 and, therefore, the determination of whether the delivery occurred before or after diagnosis would be not accurate for deliveries in 2009 and 2010.

For hospitalization rate analysis, only pregnancies started at least 40 weeks after the first MS diagnosis and terminated not later than in March of 2019 were included, providing a sufficiently long time window to inspect 40 weeks before pregnancy, throughout the whole period of pregnancy and 40 weeks after pregnancy. Ultimately, 1610 deliveries were identified using these criteria.

Direct recognition of MS relapses based on the NFZ database is impossible. Therefore, we have made an attempt to identify services associated with relapses based on their type, duration and procedures reported. Most hospitalizations reported with the “G35” ICD-10 code and shorter than 1 day (hospital admission and discharge occurred on the same day) are scheduled hospitalizations of patients taking part in disease-modifying therapy (DMT) or diagnostics and should not be recognized as associated with relapses. Therefore, we focused on at least 1-day hospitalizations not reported with the DMT. In addition, hospitalizations reported with G35 and steroid injections procedure (ICD-9 codes: 99.23 and 99.239) as well as emergency department visits with G35 were counted, as this type of visit is most likely associated with an MS relapse.

To assess the level of participation in DMT during pregnancy, we identify all services reported to NFZ with specific DRG products, associated with DMT for MS patients. MS female patients with at least one service reported with one of those products in any time of pregnancy were considered as participating in DMT therapy during pregnancy.

Weekly hospitalization rate (number of hospitalizations per week) during pregnancy and the 40-week period after childbirth were calculated for each pregnancy individually and compared to the hospitalization rate of the same patient in the 40 weeks before using the Wilcoxon signed-rank test. Data preprocessing and statistical data analysis were performed in RStudio using *base* and *data.table* packages.

## 3. Results

### 3.1. Time Intervals between the First Diagnosis and Delivery

Time intervals between the date of the first MS diagnosis and the date of delivery were calculated for all 2876 deliveries. The distribution of time intervals indicates that MS is less frequently diagnosed during pregnancy (Figure 2). Although in the weeks preceding or following pregnancy MS is diagnosed approximately 3.5–4 times a week on average, during pregnancy the weekly average decreases to approximately once a week (Table 2).

### 3.2. Number and Rate of Hospitalizations

In 1 187 (73.7%) of 1 610 pregnancies of patients after MS diagnosis, at least one hospitalization with G35 occurred during the observed time window. For 505 of them (31.4%), at least one hospitalization with G35 reported without a DMT product was observed.

The number of hospitalizations without any DMT-products reported dropped by approximately 80% during pregnancy and returned to the pre-pregnancy level soon after delivery (Table 3, Figure 3). The number of hospitalizations with steroid injections also dropped by approximately 80% and was slightly decreased also in the first few weeks after delivery. The number of emergency department visits dropped by approximately 60% and returned to the pregnancy level soon after delivery.

To assess whether pregnancy reduces the likelihood of an MS relapse, the weekly rate of hospitalization in all of the observed time windows was calculated for each pregnancy individually. Subsequently, the rate of hospitalization during and after pregnancy was compared to the rate of hospitalization before pregnancy in the same patient.

The weekly rate of hospitalizations reported without DMT-products decreased from 0.0055 on average before pregnancy to 0.0010 during pregnancy (*p*-value < 0.001; Figure 4A, Table 3). The rate of hospitalizations was reduced in each trimester of pregnancy but mostly in the 3rd trimester (0.0007, *p* < 0.001). Soon after pregnancy, the rate of hospitalizations increased to 0.0053 and stayed at a similar level during all 40 weeks after pregnancy.

Significantly decreased hospitalization rates were observed both among women participating and not participating in DMT during pregnancy (Figure 5, Table 4). However, in case of women not participating in DMT during pregnancy, the rate of hospitalization between 14 and 26 weeks postpartum was slightly higher than before pregnancy (0.0068 vs. 0.0055, *p* = 0.045).

The same decrease during pregnancy is observed in the case of hospitalizations reported with steroid injections and emergency department visits (Figure 4B,C, Table 3). However, the rate of hospitalizations reported with steroid injections is significantly decreased also in the first trimester after pregnancy (0.0008 vs. 0.0013 before pregnancy, *p* = 0.008).

### 3.3. Participation of Pregnant MS Patients in the DMT

The level of participation in the DMT decreases immediately with the beginning of pregnancy. DMT with IFN-ß 1b is reported in the case of 152 patients in the first trimester and continued only in 10 patients in the second trimester and 6 patients in the third trimester (Table 5).

The most common drug taken in the second and third trimester is glatiramer acetate—the only disease-modifying drug approved by NFZ for use by pregnant women with MS. Only a few cases of taking other disease-modifying drugs in the 2nd and 3rd trimester were reported.

The analysis of all DMT-related medical services in the first trimester of pregnancy revealed that the most frequent were those related to the administration (or dispensation) of IFN-β, glatiramer acetate, dimethyl fumarate, fingolimod, and natalizumab. Then, the number of these services decreases so that in the last trimester only IFN-β, glatiramer acetate, dimethyl fumarate are used.

## 4. Discussion

The period of pregnancy is a protective state for women with regard to autoimmune diseases. The fetus is an allograft and generates immunity in the woman’s body [11]. The studies conducted did not show any negative long-term effects of pregnancy on the natural course of the disease and it was even believed that women with a history of multiple pregnancies had a lower degree of disability and needed a longer period of time to achieve a certain level of disability. It followed that pregnancy had an immunomodulating effect on the course of MS [12,13,14]. Moreover, the latest data confirm that pregnancy reduces the risk of relapse of disease [15].

There used to be a view that patients should be discouraged from getting pregnant. Currently, it is known that pregnancy has an immunomodulatory effect on the course of MS and does not worsen the prognosis [16]. However, there is no evidence that pregnancy can affect the long-term progression of MS [15].

One of the Polish studies assessing the reproductive health of MS patients revealed that 53.01% of MS patients declared their interest in motherhood. Patients interested in pregnancy were significantly younger (*p* < 0.01), often nulliparous (*p* < 0.001), had a lower EDSS score (*p* < 0.006) and a smaller motor deficit (*p* < 0.001) [17]. The analysis showed that the degree of disability and the age of women are important in making a decision about pregnancy.

Our study revealed, that the number of all types of hospitalizations reported without DMT products decreased by approximately 80% during pregnancy. Interestingly, hospitalizations of at least one day without DMT-products and hospitalizations in emergency departments return to the level observed before pregnancy in the first weeks after delivery, while hospitalizations with steroid injections also occurred less frequently in the first weeks after delivery. Nevertheless, the latter hospitalizations were associated with the highest probability of disease relapse among the group of patients analyzed. This may indirectly suggest that the post-pregnancy time among women from the Polish population was not associated with an increased risk of disease recurrence compared to the period before pregnancy.

This was in contradiction with some data from the literature. Some authors indicated that the frequency of MS relapses tends to be reduced in late pregnancy but increases in the postpartum period. The causes of increased postpartum activity were not entirely clear but factors, such as the sharp drop in estrogen levels immediately after delivery and loss of the immunosuppressive state of pregnancy, were likely to play a role [18].

Confavreux et al., examined 254 women with MS. The study was conducted as a multicentre study in 12 European countries. Recruitment was carried out between 4 and 36 weeks of gestation and follow-up continued for one year postpartum. During pregnancy, the number of relapses decreased and reached 0.2 in the third trimester. In the next three months after giving birth, the index was already 1.2. Moreover, it turned out that women with higher disease activity in the year preceding pregnancy and during pregnancy, and with a high degree of motor disability, have a higher risk of recurrence within three months after delivery [19].

In our study, the weekly rate of hospitalizations reported without DMT-products decreased from 0.0055 on average before pregnancy to 0.0010 during pregnancy, which may indirectly indicate a decrease in relapsing activity during pregnancy. In addition, we also observed a decrease in the number of first MS diagnoses received during pregnancy. Among 344 MS patients diagnosed during the 2.3-year observation window, only 11% received MS diagnosis during pregnancy, compared with 41% and 48% who received a diagnosis before or after pregnancy over a period of similar duration. Since a first MS diagnosis is often made in consequence of relapse, this may also be an indirect indication of decreased relapsing activity during pregnancy.

Recent studies showed that patients with MS did not develop relapses after delivery because they were previously included in drug programs and received pre-pregnancy treatment and they quickly return to treatment after delivery. Before the onset of the DMT, the course of the disease, a postpartum exacerbation was described as the natural course of the disease [20,21].

Hughes et al., revealed that the use of treatment in the two years before pregnancy reduced the risk of postpartum MS recurrence by 45% and the authors suggested that the treatment should be continued until pregnancy [22].

Hospitalization indicated the possibility of clinically overt disease progression and, as one study showed, in 21 out of 28 (75%) postpartum patients, MS disease activity presented through a relapse or MRI T2 activity [23].

The recent reports emphasize the importance of MRI imaging in predicting the course of the disease during and after pregnancy. It seems that both the MRI activity before pregnancy and the Peripartum MRI activity may play a role here [24,25].

Our study revealed, that the participation of pregnant women in DMT fell sharply in the first weeks of pregnancy. The smallest decrease was observed for glatiramer acetate, which is the most commonly reported drug in the second and third trimesters. It is the only DMT approved by the National Health Fund for use during pregnancy. To sum up, the share of MS patients in DMT during 2nd and 3rd trimester of pregnancy is insignificant.

We observed a decrease in the number of hospitalizations during pregnancy both among women participating in DMT at any time of pregnancy and those not participating in DMT. In case of the latter group, the level of hospitalization seems to be temporarily slightly elevated for a period of several weeks after delivery. This finding may confirm the positive effect of DMT during pregnancy on relapsing activity postpartum and partly explained the discrepancy of results between our study and some older studies. It should be noted that patients recognized by us as participating in DMT during pregnancy in major part have started therapy sometime before pregnancy.

Among women with active disease, the decision to start immunomodulatory therapy should not be delayed, and pregnancy is not recommended during this period. According to the current ECTRIMS/EAN guidelines (2018), when a patient with MS plans to become pregnant, the patient’s clinical status should first be established (i.e., no relapses for at least 1–2 years), and then pregnancy should be attempted [26]. If there is a high risk of disease reactivation during pregnancy, treatment with IFN beta or GA may be considered until pregnancy is confirmed [26]. Continuation of therapy with GA or with beta interferons may be considered in selected cases (in active disease), if the potential benefit outweighs the risk associated with the therapy [26].

The drug administered during pregnancy should demonstrate a high safety profile for the fetus. As the drug with the highest teratogenic risk, teriflunomide did not appear in any trimester. Some drugs are relatively safe during pregnancy and breastfeeding, while others may require a break from 6 months to 2 years before conception [27]. Interferon-beta was associated with an increased risk of spontaneous abortion and lower birth weight [28], but other researchers concluded that the drug is safe for pregnancy outcomes [29,30,31,32,33]. However, a newer study did not confirm the risks associated with INF-beta treatment. Hellwig et al., proved in their study that exposure to IFN-β before conception and/or during pregnancy does not increase the incidence of birth defects or spontaneous abortions [34].

There are many reports of the use of glatiramer acetate during pregnancy and so far there is no evidence of an increased risk of fetal malformations [35]. There are also reports of successful pregnancies during treatment with natalizumab and mitoxantrone [36,37].

Despite abundant evidence of a positive effect of pregnancy on MS, there is a group of patients with a very active form of MS, where the risk of relapse is high, even during pregnancy. These women may give up motherhood because of fear for their health condition.

Although pregnancy has an immunomodulatory effect on MS, motherhood should not be recommended in this highly active and aggressive MS [38,39]. Obtaining a good response to immunomodulating treatment within 2 years is the right moment to make a decision about pregnancy [40].

A new issue that is being intensively researched is the safety of using highly effective drugs during pregnancy. For example, Natalizumab can be continued until the 34th week of pregnancy in patients requiring long-term treatment [41]. Another safe monoclonal antibody may be rituximab [42].

To sum up, in accordance with the Polish guidelines of 2020, the continuation of natalizumab treatment during pregnancy should be considered after careful discussion with patients of the potential implications (e.g., hematological disorders of the newborn) in patients with high disease activity who are treated with natalizumab and intend to become pregnant when stabilization disease is not achieved [26]. In the aggressive course of the disease, consideration should be given to such therapy, the discontinuation of which will not be associated with the risk of reactivation (cladribine or alemtuzumab). During such treatment, patients can plan pregnancy four months after the last dose of alemtuzumab and six months after the last dose of cladribine. Women who experience relapses during pregnancy can be successfully and safely treated with intravenous immunoglobulins or glucocorticoids [26].

The main limitation of our study is the indirect identification of MS relapses. Due to the lack of more precise data in administrative databases, we have been identifying MS-related hospitalizations not reported as DMT-product as a proxy for MS relapses. It should be noted that, in fact, not all of these hospitalizations must have been associated with relapses (the hospitalizations reported with steroid injections and emergency department visits are most likely). On the other hand, the method used by us did not allow the detection of mild relapses that do not require hospitalization.

## 5. Conclusions

MS often affects women of childbearing age. The study showed that each clinical case should be considered individually. The patient should be informed about the possibility of relapse after pregnancy. However, pregnancy was naturally associated with a 70% reduction in the frequency of relapses in the third trimester. This effectiveness is approximately equal to the most effective disease-modifying MS treatments [11,18].

The data from the report indicated that there was a high probability that the frequency of relapses could be similar to that before pregnancy. This may be the result of previous therapy that did not lead to disease progression as well as a quick return to treatment after pregnancy. The personalization of treatment in the context of family planning remains a key issue.

## Figures and Tables

**Figure 1 jcm-11-05615-f001:**
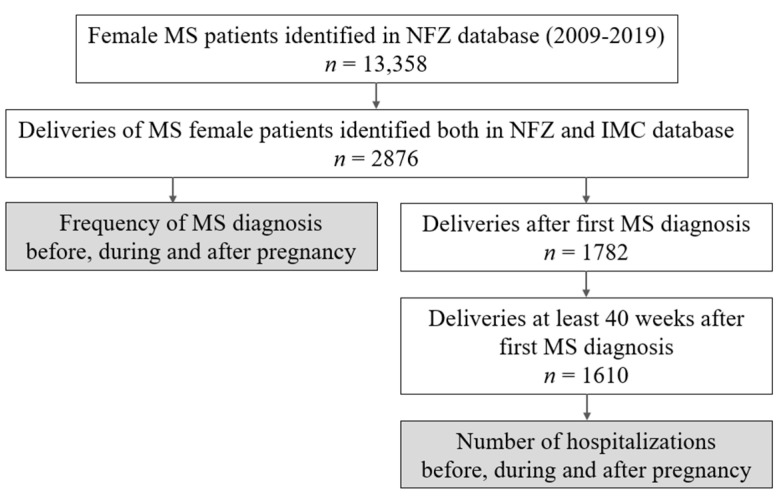
Scheme of study samples selection.

**Figure 2 jcm-11-05615-f002:**
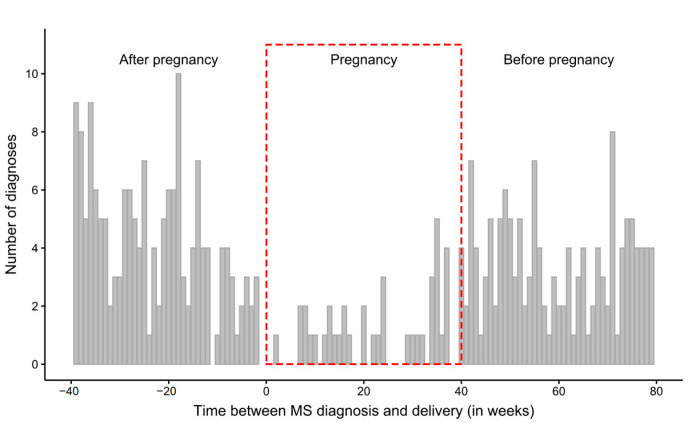
Distribution of time intervals between the first MS diagnosis and delivery. The observation window covers 40 weeks before pregnancy, during pregnancy (marked with a red frame) and 40 weeks postpartum. Positive interval (right side of the chart) values indicate that the first MS diagnosis occurred before delivery, while negative ones (left side of the chart) indicate that delivery occurred before the first MS diagnosis. The time interval between 0 and approximately 40 indicates that the first MS diagnosis occurred during pregnancy.

**Figure 3 jcm-11-05615-f003:**
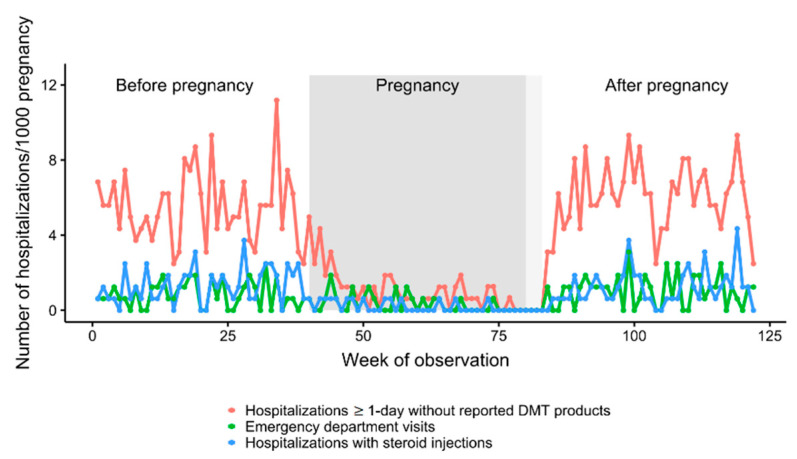
Weekly sums of hospitalizations of a given type per 1000 of observed pregnancies. The observation window covers 40 weeks before pregnancy, the entire gestation period (highlighted in gray) and 40 weeks postpartum.

**Figure 4 jcm-11-05615-f004:**
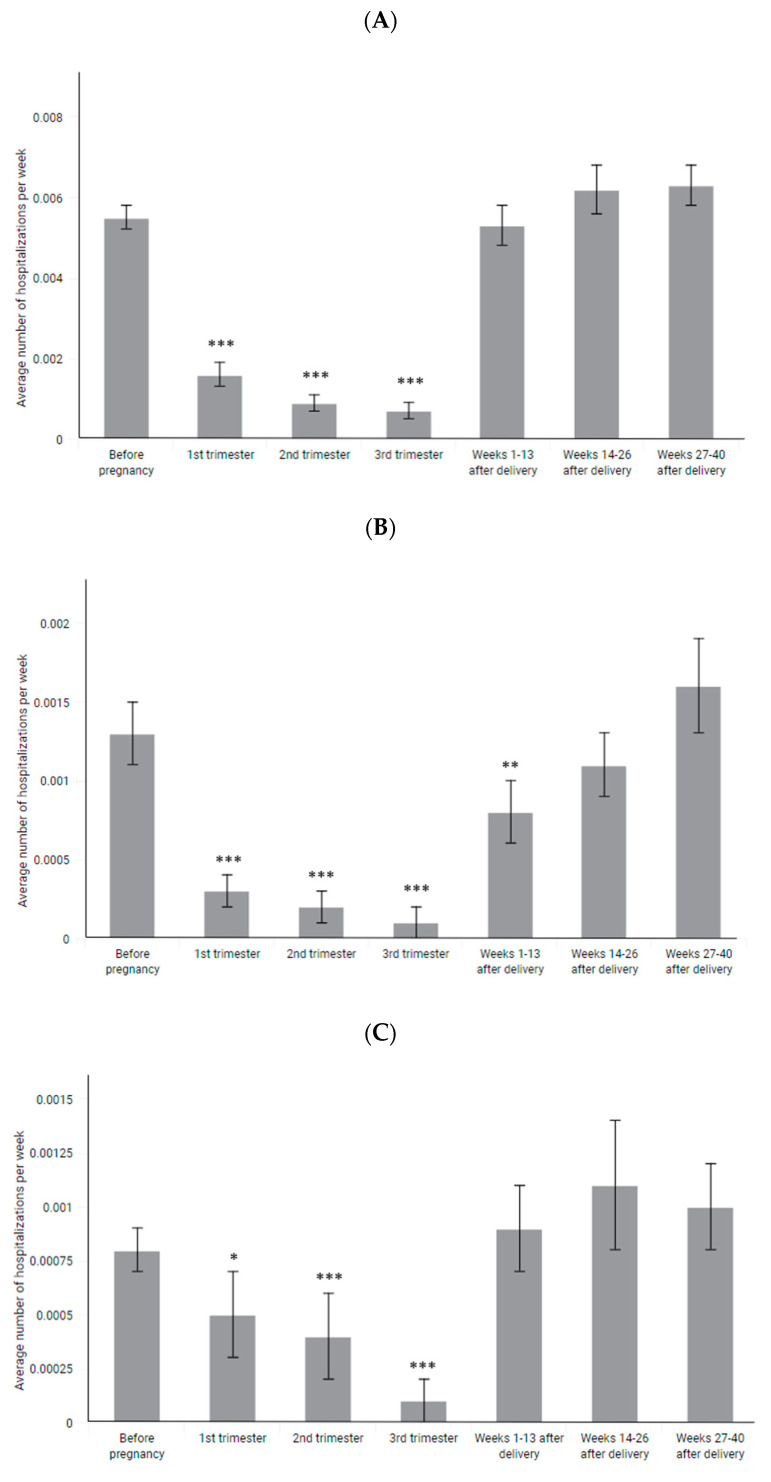
Weekly rate of (**A**) at least 1-day hospitalizations without reported DMT products, (**B**) at least 1-day hospitalizations reported with steroid injection procedure, (**C**) emergency department visits, expressed as means with SEM. *–*p*-value < 0.05, **–*p*-value < 0.01, ***–*p*-value < 0.001.

**Figure 5 jcm-11-05615-f005:**
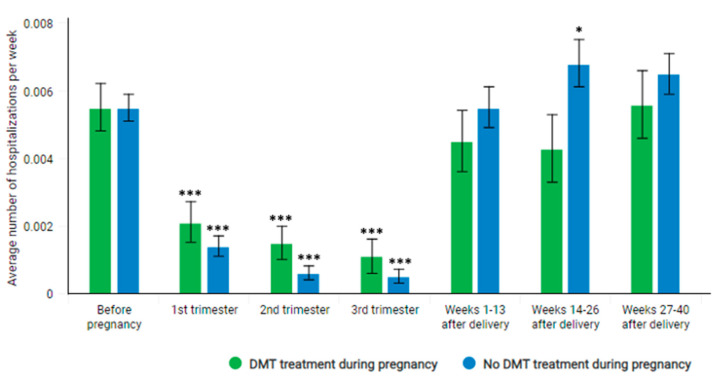
Weekly rate of at least 1-day hospitalizations without reported DMT products among women who participated or not participated in DMT during pregnancy. Rates in each period are compared with the reference period before pregnancy for each group separately. *–*p*-value < 0.05, ***–*p*-value < 0.001.

**Table 1 jcm-11-05615-t001:** Characteristics of deliveries of MS patients (*n* = 2876).

Characteristic	Mean ± SD (Min–Max)
Age at the first MS diagnosis	28.9 ± 5.4 (14–52)
Age at delivery	30.7 ± 4.7 (17–46)
Duration of pregnancy (in weeks)	38.6 ± 1.9 (23–47) *
% of multiple births	2.5
% of Ceaserian sections	56.3

*—missing data about duration for 2.3% of pregnancies.

**Table 2 jcm-11-05615-t002:** Number and weekly mean of the first MS diagnosis in the 40 weeks before, during and after pregnancy.

Time	Sum of Diagnoses	Weekly Average
40 weeks preceding pregnancy	141	3.52
40–42 weeks of pregnancy	39	0.98
40 weeks following pregnancy	164	4.10

**Table 3 jcm-11-05615-t003:** Number of hospitalizations before, during and after pregnancy with the standard error of the mean. The weekly rate of hospitalizations for each pregnancy during and after pregnancy was compared to the rate before pregnancy using Wilcoxon tests for pairs.

Period	N	Sum of Hospitalizations	Number of Pregnancies for Which at Least 1 Hospitalization Occurred	Rate of Hospitalization per Week for Each Pregnancy Individually		
				Mean	SEM	*p*-Value
** *All ≥ 1 day hospitalizations without reported DMT-products* **						
Before pregnancy (40−1. week before pregnancy)	1610	354	283	0.0055	0.0003	Reference
Pregnancy	1610	64	54	0.0010	0.0002	<0.0001
*week 1–13*	*1610*	*33*	*31*	*0.0016*	*0.0003*	*<0.0001*
*week 14–26*	*1610−1605*	*18*	*18*	*0.0009*	*0.0002*	*<0.0001*
*week 27–42*	*1610−8*	*13*	*12*	*0.0007*	*0.0002*	*<0.0001*
After pregnancy	1610	381	300	0.0059	0.0004	0.3044
*week 1–13*	*1610*	*110*	*104*	*0.0053*	*0.0005*	*0.6570*
*week 14–26*	*1610*	*129*	*117*	*0.0062*	*0.0006*	*0.2962*
*week 27–40*	*1610*	*142*	*133*	*0.0063*	*0.0005*	*0.5023*
** *Hospitalizations reported with steroid injection procedures* **						
Before pregnancy (40 *−* 1. week before pregnancy)	1610	85	77	0.0013	0.0002	Reference
Pregnancy	1610	13	11	0.0002	0.0001	<0.0001
*week 1–13*	*1610*	*7*	*7*	*0.0003*	*0.0001*	*<0.0001*
*week 14–26*	*1610−1605*	*4*	*4*	*0.0002*	*0.0001*	*<0.0001*
*week 27–42*	*1610−8*	*2*	*1*	*0.0001*	*0.0001*	*<0.0001*
After pregnancy	1610	76	64	0.0012	0.0002	0.3855
*week 1–13*	*1610*	*17*	*17*	*0.0008*	*0.0002*	*0.0083*
*week 14–26*	*1610*	*23*	*23*	*0.0011*	*0.0002*	*0.1696*
*week 27–40*	*1610*	*36*	*33*	*0.0016*	*0.0003*	*0.8422*
** *Hospitalizations in emergency department* **						
Before pregnancy (week 1–40)	1610	54	42	0.0008	0.0001	Ref.
Pregnancy	1610	21	17	0.0003	0.0001	0.0373
*week 1–13*	*1610*	*11*	*10*	*0.0005*	*0.0002*	*0.0351*
*week 14–26*	*1610−1605*	*8*	*6*	*0.0004*	*0.0002*	*0.0008*
*week 27–42*	*1610−8*	*2*	*2*	*0.0001*	*0.0001*	*<0.0001*
After pregnancy	1610	65	54	0.0010	0.0001	0.264
*week 1–13*	*1610*	*19*	*16*	*0.0009*	*0.0002*	*0.7534*
*week 14–26*	*1610*	*24*	*21*	*0.0011*	*0.0003*	*0.4583*
*week 27–40*	*1610*	*22*	*22*	*0.0010*	*0.0002*	*0.6805*

**Table 4 jcm-11-05615-t004:** Number of hospitalizations before, during and after pregnancy with the standard error of the mean. The weekly rate of hospitalizations for each pregnancy during and after pregnancy was compared to the rate before pregnancy using Wilcoxon tests for pairs, separately for groups of patients participating or not participating in DMT during pregnancy.

Period	N	Sum of Hospitalizations	Number of Pregnancies for Which at Least 1 Hospitalization Occurred	Rate of Hospitalization per Week for Each Pregnancy Individually		
				Mean	SEM	*p*-Value
** *Patients participating in DMT during pregnancy* **						
Before pregnancy (40−1. week before pregnancy)	410	91	74	0.0055	0.0007	Reference
Pregnancy	410	25	21	0.0016	0.0004	<0.0001
*week 1–13*	*410*	*11*	*11*	*0.0021*	*0.0006*	*<0.0001*
*week 14–26*	*410*	*8*	*8*	*0.0015*	*0.0005*	*<0.0001*
*week 27–42*	*410−2*	*6*	*5*	*0.0011*	*0.0005*	*<0.0001*
After pregnancy	410	79	68	0.0048	0.0006	0.3897
*week 1–13*	*410*	*24*	*24*	*0.0045*	*0.0009*	*0.2804*
*week 14–26*	*410*	*23*	*21*	*0.0043*	*0.0010*	*0.1056*
*week 27–40*	*410*	*32*	*32*	*0.0056*	*0.0010*	*0.7486*
** *Patients not participating in DMT during pregnancy* **						
Before pregnancy (40−1. week before pregnancy)	1200	263	209	0.0055	0.0004	Reference
Pregnancy	1200	39	33	0.0009	0.0002	*<0.0001*
*week 1–13*	*1200*	*22*	*20*	*0.0014*	*0.0003*	*<0.0001*
*week 14–26*	*1200−1195*	*10*	*10*	*0.0006*	*0.0002*	*<0.0001*
*week 27–42*	*1194−6*	*7*	*7*	*0.0005*	*0.0002*	*<0.0001*
After pregnancy	1200	302	231	0.0063	0.0004	0.095
*week 1–13*	1200	*86*	*80*	*0.0055*	*0.0006*	*0.9295*
*week 14–26*	1200	*106*	*96*	*0.0068*	*0.0007*	*0.0448*
*week 27–40*	1200	*110*	*104*	*0.0065*	*0.0006*	*0.3486*

**Table 5 jcm-11-05615-t005:** Participation of pregnant MS patients in the DMT by type of drug.

Drug Used	Number of Pregnancies for Which at Least One Visit Reported with DMT (with Given Drug) Was Observed		
	Week 1–13 (1st Trimester)	Week 14–26 (2nd Trimester)	Week 27–42 (3rd Trimester)
IFN-ß 1b	152	10	6
IFN-ß 1a	130	7	2
Glatiramer acetate	93	25	24
Dimethyl fumarate	46	2	3
Fingolimod	14	2	0
Natalizumab	6	1	0

## Data Availability

The data in this study were obtained with the permission of the Ministry of Health of the Republic of Poland from electronic databases of the National Health Fund (NFZ). Datasets are not publicly available because they contain sensitive data at an individual level. Aggregated data may be requested from the Department of Analyses and Strategies in the Ministry of Health in accordance with the provisions on access to public information (a justification for the public interest is required).

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
