# Peer review of "The Rate of Hospitalization of Pregnant Women with Multiple Sclerosis in Poland"

_jcm, 2022, doi:10.3390/jcm11195615_

Round 1

Reviewer 1 Report

In this nationwide study, the authors investigated the rate of hospitalization 40 weeks before, after, and during pregnancy. The date of the live birth procedure was used to estimate the date of conception and the pregnancy periods. They found decreased number of hospitalization and relapse within pregnancy compared to the before pregnancy. The rates returned to the pre-pregnancy after delivering. The objective of this study is important, however the novelty of the study in limited.

There are some comments:

-Please explain the main aims of the study in the last paragraph of introduction

-Please declare the novelty of the study in Introduction section

-Is NFZ database previously validated for MS diagnosis?

-How authors differentiated MS-related hospitalization and non-MS –related hospitalization? Patients may visit hospital for infectious diseases

-It is possible that patients receive corticosteroid therapy for reason rather than relapse, such as arthritis rheumatoid

- Authors used a relapse detection algorithm to identify relapse in MS patients. These algorithm is not suitable to identify mild relapse since they do not need hospitalization and steroids. Moreover, some patients may receive therapy for relapses in the outpatient or use home care.

-It is suggested to consult with a statistician. Signed-rank test shows subpar power compared to GEE method

-Please provide per-review articles for references 34-40  

Reviewer 2 Report

Summary: 

This manuscript reports the results of a study, in which the authors investigate the hospitalization rates of women with MS before during and after pregnancy as a proxy for relapse rate. A number of previous studies have already shown that the relapse rate decreases during pregnancies and the authors should report and discuss the findings of these previous studies more clearly. The most interesting finding might be that in contrast to the results of previous studies in this cohort the authors did not observe an increase of the relapse rate postpartum. This should be highlighted more and should also be investigated in more detail – e.g. with regards to the relationship of post-partum relapses and DMT use before and during pregnancies.  

Please find my comments and questions below. 

Introduction 

-        Lines 49-52: The statement that the course of MS may be alleviated during pregnancies because the fetus contains paternal proteins needs to be supported by evidence (references) and explained well. There might well be other explanations for the reduced relapse rate of MS during pregnancies (e.g. hormonal changes) that should be mentioned. 

-        Lines 49-52: A number of studies already investigated the occurrence of relapses during and after pregnancy – these should be mentioned here (e.g. Confavreux et al. NEJM 1998, Houtchens et al., Neurology 2018).  

-        Line 59: In my opinion, the sentence “In one study, 71.2% were the causes related to MS that made parenting difficult.“ is misleading as it implies that in 71.2% of cases women are having difficulties becoming parents due to MS-related biological causes. However, in the references study the authors state that the main MS-related reason for not becoming pregnant “was the perception that symptoms would interfere with parenting (71.2%)”. 

-        Line 61-62: I believe that the authors should add references that support this point.  

Material and Methods

-        Lines 87-88: It seems that women with MS without DMT were excluded from this study. This might introduce bias as it potentially excludes women with mild disease from the study. I believe the authors should describe the reason for this restriction and discuss possible impact on the study results. 

-        Table 1: The percentage of ceaserian section (56%) seems quite high. Is this specific for this MS cohort or is this a typical percentage in Poland?

-         

Results: 

-        Regarding DMT during pregnancy – if I understood it correctly, DMT use can only be inferred from hospitalization with a record of DMT. However, I believe that patients under treatment with interferons or glatiramer acetate do not need to be hospizalized for treatment. If that is the case also in this cohort, DMT usage during pregnancy could be severly underestimated. Could the authors comment on this? 

-         

Discussion: 

-        I believe that the authors should interpret and discuss the findings regarding time point of MS diagnosis with regards to pregnancies (Figure 2 and Table 2). 

-        Lines 239-241: Could the authors please explain what a number of relapses of 0.2 means? Relapses per pregnancy or per person or relative to pre-pregnancy relapse rate?

-        Would it be possible to investigate whether women without DMT during the pregnancies have a higher hospitalization rate after delivery as compared to women who were treated during pregnancy? This could support the assumption that DMT treatment explains the differences regarding post-delivery relapse activity between this and previous studies.

-        Lines 270-271: See my comment under the results section regarding DMT use during pregnancy. 

-        Lines 272-273: I don’t believe this statement to be correct in many cases. Women with MS under DMT who are planning to get pregnant will probably perform pregnancy tests rather frequently and will know about their pregnancy early. 

-        Lines 275-277 (“In addition, the natural course of MS during pregnancy (associated with a low annual relapse rate) makes the need for the DMT during pregnancy questionable.“). It should be discussed that each relapse can lead to significance (long-term) disability - I find this statement too strong. 

-        Lines 311-314: I am not convinced that there is enough evidence that support the statement that there is a relationship between the aggressiveness of MS and impairment of fertility. The authors do not name any references that support this statement. Furthermore, the relation between this statement and the following sentence is not clear to me. 

-        The part about the risks of different DMT with regards to pregnancies is a bit confusing and could be restructured. 

Round 2

Reviewer 1 Report

Authors have addressed all my comments